Contrasting effects of ocean acidification on tropical fleshy and calcareous algae

Johnson Maggie Dorothy mdjohnson@ucsd.edu
Price Nichole N.
Smith Jennifer E.
Scripps Institution of Oceanography , La Jolla, CA , USA
Toonen Robert
Electronic publication date: 2014 May 27
Publication date: 2014
Volume: 2
Electronic Location ID: e411
Received 2014 Jan 29; Accepted 2014 May 12
Copyright: © 2014 Johnson et al.
Copyright year: 2014
Copyright holder: Johnson et al.
License: This is an open access article distributed under the terms of the Creative Commons Attribution License, which permits unrestricted use, distribution, reproduction and adaptation in any medium and for any purpose provided that it is properly attributed. For attribution, the original author(s), title, publication source (PeerJ) and either DOI or URL of the article must be cited.
License URL: https://creativecommons.org/licenses/by/4.0/

Keywords: Calcification, Crustose coralline algae, Halimeda, Macroalgae, Ocean acidification, Photosynthesis, Calcareous algae

Funding: Gordon and Betty Moore Foundation Scripps Family Foundation Rhodes and Bohn Families Funding was generously provided by grants from the Gordon and Betty Moore Foundation, Scripps Family Foundation, and the Rhodes and Bohn Families to JE Smith. The funders had no role in study design, data collection and analysis, decision to publish, or preparation of the manuscript.

==============================
Despite the heightened awareness of ocean acidification (OA) effects on marine organisms, few studies empirically juxtapose biological responses to CO2 manipulations across functionally distinct primary producers, particularly benthic algae. Algal responses to OA may vary because increasing CO2 has the potential to fertilize photosynthesis but impair biomineralization. Using a series of repeated experiments on Palmyra Atoll, simulated OA effects were tested across a suite of ecologically important coral reef algae, including five fleshy and six calcareous species. Growth, calcification and photophysiology were measured for each species independently and metrics were combined from each experiment using a meta-analysis to examine overall trends across functional groups categorized as fleshy, upright calcareous, and crustose coralline algae (CCA). The magnitude of the effect of OA on algal growth response varied by species, but the direction was consistent within functional groups. Exposure to OA conditions generally enhanced growth in fleshy macroalgae, reduced net calcification in upright calcareous algae, and caused net dissolution in CCA. Additionally, three of the five fleshy seaweeds tested became reproductive upon exposure to OA conditions. There was no consistent effect of OA on algal photophysiology. Our study provides experimental evidence to support the hypothesis that OA will reduce the ability of calcareous algae to biomineralize. Further, we show that CO2 enrichment either will stimulate population or somatic growth in some species of fleshy macroalgae. Thus, our results suggest that projected OA conditions may favor non-calcifying algae and influence the relative dominance of fleshy macroalgae on reefs, perpetuating or exacerbating existing shifts in reef community structure.

Introduction

Changes in ocean chemistry associated with anthropogenic carbon dioxide (pCO2) emissions, a process known as ocean acidification (OA) (Kleypas et al., 1999; Orr et al., 2005), have raised widespread concern for the survival and persistence of marine biota (Kleypas et al., 1999; Hoegh-Guldberg et al., 2007). Identifying the groups of organisms that will be susceptible to rapid OA versus those that may be resistant has prompted numerous studies (Ries, Cohen & McCorkle, 2009; Kroeker et al., 2010; Kroeker et al., 2013). To date, research has focused on understanding how reductions in the saturation state (Ω) of calcium carbonate (CaCO3) and seawater pH associated with OA will impact the growth and physiology of commercially important calcifying organisms or entire ecosystems, such as coral reefs, that build carbonate platforms (Kleypas et al., 1999; Andersson & Gledhill, 2013). However, examination of a wider taxonomic representation, including those that calcify and those that do not, within and across ecosystems is critical to developing ecological predictions of community-level responses to OA.

The changes in the carbonate system have important implications for marine calcifiers, namely that OA may inhibit the ability of these species to grow, develop, reproduce and sustain themselves within a community, although plasticity in organismal responses indicates that some species may have wider tolerance limits (Doney et al., 2009; Kroeker et al., 2010; Kroeker et al., 2013; Johnson, Moriarty & Carpenter, 2014). Mounting evidence from coral reefs suggests that decreasing carbonate saturation (Ω) has negative effects on calcification (Langdon & Atkinson, 2005; Doney et al., 2009; Andersson & Gledhill, 2013), reproductive success (Albright, 2011), and competitive strength (Diaz-Pulido et al., 2011) of scleractinian corals. However, less attention has been given to the response of tropical marine primary producers to rising oceanic CO2, particularly fleshy and calcareous benthic macroalgae which are also among the most dominant constituents of the coral reefs benthos.

The future trajectory of coral reefs may be influenced by concurrent effects of OA on both fleshy and calcified algae (reviewed in Koch et al., 2013), which serve key functional roles in reef systems in addition to competing with corals for space and resources. Calcareous algae contribute to framework development and some are active reef builders that account for up to 90% of living benthic cover on reefs (Tribollet & Payri, 2001). Crustose coralline algae (CCA) serve important ecological functions on reefs by contributing to primary production and carbonate production (Chisholm, 2003), producing settlement cues for coral larvae (Harrington et al., 2004; Price, 2010), and maintaining structural integrity of the framework by acting as reef cement (Camoin & Montaggioni, 1994). Calcareous green algae, such as Halimeda spp., are a major source of primary production and CaCO3 (Rees et al., 2007) due to their fast growth and turnover rates (Smith et al., 2004), and are a preferred food source for many coral reef fishes (Mantyka & Bellwood, 2007; Hamilton et al., 2014). Fleshy macroalgae include a highly diverse group of seaweed species that act as a source of food for higher trophic levels and directly compete with corals for space (McCook, Jompa & Diaz-Pulido, 2001) on the reef benthos. Some fleshy macroalgae produce toxic allelochemicals which can kill corals upon contact (Rasher et al., 2012) while others may transmit coral disease (Nugues et al., 2004) or affect microbial assemblages associated with the coral holobiont via release of dissolved organic carbon (Smith et al., 2006; Haas et al., 2013; Nelson et al., 2013). Furthermore, the relative balance of calcifiers to fleshy macroalgae is important for reef resilience (Hughes et al., 2010). Increased cover of fleshy macroalgae, associated with anthropogenic disturbances such as poor water quality (Fabricius, 2005) and overfishing, is often used as an indicator of deteriorating reef health (Hughes, 1994). Given the important roles that calcareous and fleshy algae serve in the structure and function of coral reef ecosystems, it is critical to identify the potential differential effects of OA on these functionally distinct groups.

Increased CO2 has the potential to have disparate effects on physiological processes for calcareous and fleshy algae, namely on photosynthesis and biomineralization. In terrestrial systems, rising atmospheric CO2 can fertilize primary producers and enhance production (Ainsworth & Long, 2005), but in marine ecosystems, photosynthesizers have access to other relatively abundant carbon species, such as bicarbonate (HCO3−), that can be used for photosynthesis. The potential for CO2 fertilization of marine primary producers is likely contingent on species–specific mechanisms of carbon acquisition, influenced by the activity of carbon concentrating mechanisms (CCMs) (Giordano, Beardall & Raven, 2005; Raven et al., 2011; Koch et al., 2013). Laboratory manipulations and field studies from temperate and Mediterranean ecosystems (Hall-Spencer et al., 2008; Porzio, Buia & Hall-Spencer, 2011) suggest that OA may enhance carbon fixation (Kroeker et al., 2010; Cornwall et al., 2012; Kroeker et al., 2013) and photosynthesis in fleshy algae resulting in increases in algal growth rates (Gao et al., 1991; Kubler, Johnston & Raven, 1999; Cornwall et al., 2012). However, variations in interspecific responses may depend on the extent to which a species is presently carbon-limited (Harley et al., 2012; Koch et al., 2013). The photosynthetic response of seaweeds to OA is poorly understood in part because data on the presence, absence, or activity level of CCMs is often lacking for many tropical species (Hurd et al., 2009; Raven et al., 2011). Although much of the literature on OA effects on marine algae has shown that CO2 enrichment enhances photosynthesis in phytoplankton and phanerograms (Riebesell et al., 1997; Palacios & Zimmerman, 2007; Gattuso & Hansson, 2011), the photosynthetic response of seaweeds to OA has been highly variable across experiments (Koch et al., 2013) and sometimes negative for calcified species (Anthony et al., 2008; Sinutok et al., 2011; Sinutok et al., 2012).

Conversely, OA effects on skeletal production in calcareous algae have been studied in more detail and changes in carbonate chemistry (i.e., lower pH, lower carbonate availability, and decreased CaCO3 saturation state) have been shown to inhibit calcification in many species. The effects of OA on calcification in marine organisms may be influenced by the ability for a species to control carbonate chemistry at the intracellular or extracellular site of calcification (Ries, Cohen & McCorkle, 2009). A decrease in Ω in the external environment associated with OA could make biogenic CaCO3crystal precipitation more difficult. When Ω decreases below the saturation horizon (<1) CaCO3 dissolution is thermodynamically favored (Milliman et al., 1999). This saturation horizon is influenced by temperature, pressure, and mineralogy (Feely et al., 2004; Orr et al., 2005), and net dissolution and calcification can occur both above and below this threshold, respectively, depending on the organism and the environment (Milliman et al., 1999). CCA may be some of the most sensitive calcifiers to OA because they secrete high Mg-calcite, the most soluble polymorph of CaCO3 (Morse, Andersson & Mackenzie, 2006; Andersson, Mackenzie & Bates, 2008). Other studies have found that OA decreases CCA calcification (Semesi, Kangwe & Bjork, 2009; Johnson & Carpenter, 2012; Comeau, Carpenter & Edmunds, 2012; Comeau et al., 2013; Johnson, Moriarty & Carpenter, 2014), structural integrity (Ragazzola et al., 2012) and increases mortality and that these effects that may be exacerbated by warming temperatures (Anthony et al., 2008; Martin & Gattuso, 2009; Diaz-Pulido et al., 2012; Martin et al., 2013) or increased solar UV radiation (Gao & Zheng, 2010). The articulated calcareous green algae Halimeda spp. have been shown to be both sensitive (Sinutok et al., 2011; Sinutok et al., 2012) and insensitive to OA (Comeau et al., 2013), where the direction and magnitude of the response of Halimeda to OA varies among species (Ries, Cohen & McCorkle, 2009; Price et al., 2011; Comeau et al., 2013). Negative effects of OA on calcification of CCA and Halimeda spp. may have serious implications for carbonate production and framework stability on coral reefs because they are often common members of ‘intact’ benthic reef communities (Sandin et al., 2008; Williams et al., 2013).

The primary objective of our study was to determine if there are consistent, differential responses of fleshy and calcareous tropical marine algae to OA using parallel, replicated experimental manipulations. On Palmyra Atoll in the northern Line Islands, five common species of fleshy algae and six species of calcareous algae were exposed to CO2 levels expected by the year 2100 under a business-as-usual carbon emissions scenario (Meinshausen et al., 2011). In particular, the hypotheses tested were that, even with variation in species–specific physiological responses, elevated CO2, (1) reduces net calcification across calcareous algae but, (2) stimulates growth of fleshy algae by enhancing photosynthesis. This study provides one of the first efforts to quantify OA effects on multiple species of both calcareous and fleshy algae from a coral reef environment, and provides insight into the effects of OA on a suite of algae that are important in the structure and function of coral reefs.

Materials and Methods

Study site and species

All experiments were conducted on Palmyra Atoll in the Northern Line Islands, central Pacific, in the recently established Pacific Remote Island Areas Marine National Monument (PRIAMNM) protected by the US Fish and Wildlife Refuge. Due to its isolation (∼1,700 km south-southwest of Hawaii) and lack of permanent human residence, Palmyra’s coral reefs are considered relatively healthy and are dominated by reef builders (Sandin et al., 2008; Williams et al., 2013). The remote nature of the field station limits research excursions to a few weeks at a time. Due to the absence of potentially confounding local anthropogenic impacts, Palmyra provides a unique setting for global change experiments.

To explore the effects of OA on different algal functional groups, eleven common species of algae were used in CO2 enrichment experiments (see Sandin et al., 2008 and Williams et al., 2013 for relative abundances). Algae were categorized into three functional groups: fleshy macroalgae (Acanthophora spicifera, Caulerpa serrulata, Dictyota bartayresiana, Hypnea pannosa, and Avrainvillea amadelpha), upright calcareous algae (Halimeda taenicola, Halimeda opuntia, Galaxaura rugosa, and Dichotomaria marginata), and crustose coralline algae (CCA: Lithophyllum prototypum, formerly Titanoderma prototypum, and Lithophyllum sp.) (Fig. 1, Table S1). Specimens were collected via SCUBA at a depth of ∼5 m from the shallow western terrace (5°53.1696′N, 162°7.5756′W), excluding L. prototypum. L. prototypum was collected at a depth of ∼10 m from the southern fore reef (5°53.7906′N, 162°7.6859′W) where the species is abundant. Except for the corallines, which were collected as free-living rhodoliths, individuals were removed at the holdfast or from rhizoids in order to minimize stress. Coralline rhodoliths were comprised of 100% live coralline cover, and no bare carbonate was exposed to the potentially corrosive conditions. Samples were cleaned carefully of epiphytes with a soft-bristled brush and allowed to acclimate for at least one day in fresh, ambient seawater.

Figure 1 Growth response of fleshy and calcareous algae to treatment conditions.

The eleven species of algae exposed to CO2 enrichment experiments on Palmyra Atoll. Algae were separated by functional group. The species of fleshy macroalgae included: (A) Acanthophora spicifera, (B) Avrainvillea amadelpha, (C) Caulerpa serrulata, (D) Dictyota bartayresiana, (E) Hypnea pannosa. The upright calcareous algae included: (F) Dichotomaria marginata, (G) Galaxaura rugosa, (H) Halimeda taenicola, (I) Halimeda opuntia, and the CCA included: (J) Lithophyllum sp., (K) Lithophyllum prototypum. (L) The mean (±SE) change in either fleshy or calcareous biomass (highlighted in gray) following exposure to either present-day ambient air controls (open circles) or predicted OA treatments (closed circles). Fleshy macroalgae are shown in brown, upright calcareous algae in green, and CCA in red. Species tested in multiple experiments were pooled across years. * Indicates a significant difference between treatments as determined by independent t-tests (results reported in Table 2).

Experimental conditions and seawater chemistry

To explore the effects of OA on growth, calcification and photophysiology of benthic algae, CO2 enrichment experiments were conducted for ∼2 weeks in July of 2010, and September of 2009, 2011, and 2012 (see Table S2 for experiments across years). Experimental aquariums (glass jars) held 700 mL of seawater collected from offshore and an individual alga (∼2 g live tissue). Seawater was treated continuously with either air or CO2 enriched gas for several hours prior to experimentation and changed (100%) every 48 h to prevent nutrient limitation and to maintain treatment conditions (sensu Price et al., 2011).

The effects of projected OA were simulated by micro-bubbling either pre-mixed air enriched with CO2 to ∼1,000 µatm into treatment aquariums (OA treatment) or ambient air into control aquariums. Clear polycarbonate lids reduced atmospheric equilibration, evaporation, and rainwater incursion. Air and CO2 enriched gas were bubbled continuously into treatment aquariums through wooden air stones that were placed at the bottom, center of experimental replicates. The continuous bubbling within a relatively small volume facilitated thorough mixing of the seawater within the jars. It was not possible to measure water flow within the contained jars however gas was adjusted to flow into experimental aquariums at a constant rate. Sample sizes varied by experiment and the availability of samples, but ranged from 4 replicates per treatment/species in 2009 to 10 in 2012 (Table S2). Additionally, aquariums without algae were maintained in all experiments to determine if algal metabolism affected carbonate chemistry and altered treatment conditions.

Aquariums were partially submerged in flow-through seawater baths under natural sunlight with shade cloth screens to simulate in situ temperature and irradiance levels at 5 m depth (Table S2). Temperature and light intensity within aquariums were monitored every 15 min with data loggers (Onset, HOBO Pendant Temperature Light/Data Logger) for the duration of the experiments. Light intensity was measured in Lux, and converted to photosynthetically active radiation (PAR) with the following conversion: 1 µmol quanta (400–700 nm) m−2 s−1 = 51.2 lux (Valiela, 1984). This conversion was validated by additional in situ PAR measurements made at the collection site, using an underwater spherical quantum sensor (LICOR, LI-193). In 2009 and 2010, oxygen (O2, polarographic electrode, ±0.2 mg L−1), temperature (±0.15 °C), salinity (±0.1 psu) and pHSW (±0.2) were monitored with a handheld meter (YSI Environmental Quatro). In 2011 and 2012, O2 (±0.01 mg L−1), temperature (±0.3 °C), and pHSW (±0.1) were measured with a Hach Lange HQ40 portable multi-parameter meter (IntelliCAL PHC101 Standard Gel Filled pH Electrode and IntelliCAL LDO101 Standard Luminescent Dissolved Oxygen LDO Optode). The pH probe was calibrated daily with certified Tris buffer (provided by Andrew Dickson, SIO). Using certified Tris buffer as a reference improved the accuracy of pH probe measurements to ±0.001. In each year, measurements were recorded from all aquariums in the evening (1800–2000) of each day (Table 1).

Table 1 Measured pH and dissolved oxygen of OA experiments on Palmyra Atoll.

The mean (±SE) measured pHSW and dissolved oxygen conditions for CO2 enrichment experiments conducted on Palmyra Atoll from 2009–2012. Measurements were conducted at the same time of day (∼2000) for the duration of the experiment in empty control (no biological material), ambient air, and high pCO2 treatments. Daily means were calculated within a species (n = 4, 2009; n = 6, 2010; n = 5, 2011; n = 10, 2012), and then averaged across days (14 days, 2009; 9 days, 2010; 17 days, 2011; 15 days, 2012). DO, dissolved oxygen; pHSW, pH seawater scale.

Treatment	Species	Temperature (°C)	DO (mg L−1)	pHSW	
2009 Experiments	
Ambient air	Control	29.31 ± 0.07	4.95 ± 0.13	8.08 ± 0.02	
	H. opuntia	29.26 ± 0.08	4.57 ± 0.16	8.03 ± 0.04	
	H. taenicola	29.29 ± 0.08	4.73 ± 0.12	7.99 ± 0.03	
Lithophyllum sp.	29.43 ± 0.03	5.17 ± 0.11	8.05 ± 0.01	
	L. prototypum	29.38 ± 0.02	5.16 ± 0.16	8.04 ± 0.02	
High pCO2	Control	29.33 ± 0.06	4.66 ± 0.26	7.68 ± 0.04	
	H. opuntia	29.23 ± 0.08	4.16 ± 0.21	7.63 ± 0.02	
	H. taenicola	29.25 ± 0.07	4.38 ± 0.19	7.62 ± 0.02	
	Lithophyllum sp.	29.41 ± 0.03	5.47 ± 0.10	7.68 ± 0.03	
	L. prototypum	29.38 ± 0.02	4.71 ± 0.21	7.65 ± 0.02	
2010 Experiments	
Ambient air	Control	29.25 ± 0.15	4.85 ± 0.05	8.06 ± 0.05	
	A. spicifera	29.22 ± 0.07	4.86 ± 0.06	8.08 ± 0.04	
	C. serrulata	28.95 ± 0.04	4.79 ± 0.13	8.09 ± 0.02	
	G. rugosa	29.25 ± 0.02	4.92 ± 0.06	8.09 ± 0.02	
	H. taenicola	29.36 ± 0.05	4.78 ± 0.07	7.98 ± 0.04	
High pCO2	Control	29.25 ± 0.05	4.70 ± 0.30	7.79 ± 0.13	
	A. spicifera	29.10 ± 0.05	4.57 ± 0.18	7.88 ± 0.05	
	C. serrulata	28.91 ± 0.03	4.30 ± 0.17	7.77 ± 0.06	
	G. rugosa	29.21 ± 0.07	4.50 ± 0.28	7.87 ± 0.06	
	H. taenicola	29.34 ± 0.06	4.69 ± 0.13	7.77 ± 0.11	
2011 Experiments	
Ambient air	Control	28.46 ± 0.21	7.98 ± 0.08	7.99 ± 0.06	
	C. serrulata	28.34 ± 0.06	7.93 ± 0.03	8.00 ± 0.02	
	D. bartayresiana	28.35 ± 0.01	7.97 ± 0.04	8.04 ± 0.02	
	H. pannosa	28.26 ± 0.18	7.68 ± 0.04	7.98 ± 0.11	
	D. marginata	28.44 ± 0.10	7.86 ± 0.06	8.05 ± 0.01	
	H. opuntia	28.87 ± 0.04	8.12 ± 0.04	7.97 ± 0.03	
	Lithophyllum sp.	28.92 ± 0.14	7.87 ± 0.05	8.03 ± 0.06	
High pCO2	Control	28.30 ± 0.16	8.02 ± 0.08	7.76 ± 0.06	
	C. serrulata	28.86 ± 0.09	7.85 ± 0.02	7.66 ± 0.03	
	D. bartayresiana	28.39 ± 0.01	7.93 ± 0.03	7.76 ± 0.02	
	H. pannosa	28.38 ± 0.04	7.68 ± 0.22	7.86 ± 0.04	
	D. marginata	28.87 ± 0.21	7.87 ± 0.05	7.80 ± 0.03	
	H. opuntia	28.87 ± 0.09	8.39 ± 0.09	7.69 ± 0.03	
	Lithophyllum sp.	28.46 ± 0.08	7.82 ± 0.07	7.74 ± 0.06	
2012 Experiments	
Ambient air	Control	28.61 ± 0.10	7.81 ± 0.02	8.11 ± 0.03	
	A. amadelpha	28.72 ± 0.18	7.84 ± 0.02	8.02 ± 0.01	
	H. taenicola	28.76 ± 0.08	7.90 ± 0.05	8.01 ± 0.03	
High pCO2	Control	28.71 ± 0.07	7.81 ± 0.03	7.85 ± 0.08	
	A. amadelpha	28.68 ± 0.07	7.87 ± 0.08	7.75 ± 0.05	
	H. taenicola	28.72 ± 0.18	7.98 ± 0.20	7.73 ± 0.06	

Discrete water samples for total alkalinity (AT) and total dissolved inorganic carbon (CT) were collected from empty aquariums (controls) and a subset of experimental aquariums from both treatment levels at multiple time points during all experiments (in 2009 only samples from empty aquariums were collected). Samples were collected by siphoning treatment water into 500 mL Corning-brand Pyrex sample bottles and fixed with 200 µL saturated HgCl2, leaving a 1% head space. Water samples were transported to Scripps Institution of Oceanography (SIO) for standard carbonate chemical analyses, (SOP, sensu Dickson, Sabine & Christian, 2007) in the lab of Dr. Andrew Dickson. AT was determined using an open-cell titrator (Metrohm Dosimat Model 665) and Metrohm potentiometric pH (SOP 3b), and CT was determined with a Single Operator Multi-parameter Metabolic Analyzer (SOMMA) coulometer (SOP 2) (Dickson, Sabine & Christian, 2007). From the measurements of AT and CT, the remaining carbonate parameters were calculated using the computer program CO2SYS (Table S3) (Pierrot, Lewis & Wallace, 2006). The average difference (±SE) between the mean measured pHSW and the mean pHSW calculated from measurements of ATand CT was 0.1 (±0.05) (n = 32).

CO2 effects on growth and calcification

Growth of fleshy algae was measured as the change in wet weight over time (to the nearest 0.01 g). Samples were spun in a salad spinner (10 revolutions) and then gently blotted dry with paper towels immediately prior to obtaining weights. Net growth and calcification were measured using the change in buoyant weight (Davies, 1989), where all calcareous species were weighed to the nearest 0.001 g while suspended (from the weigh-below on a balance) in a basket submerged in ambient seawater; a technique that works well for upright calcareous algae (Price et al., 2011). Any segments shed during the course of the experiment were weighed along with the intact thallus. Buoyant weight was converted to actual weight based on the density of seawater and the density of the respective CaCO3 polymorph. Growth and calcification rates were calculated by the change in weight over the experiment, with rates normalized to initial thallus weight and number of days in treatment conditions, expressed as change in weight per day (mg g−1 day−1).

CO2 effects on photophysiology

To assess the effect of CO2 enrichment on algal photophysiology, photosynthetic parameters were measured fluorometrically with a red Pulse Amplitude Modulated Fluorometer (PAM) (Walz). The fiber optic probe was clipped to the thallus halfway up the branch on an unepiphytized portion of tissue with the “dark leaf clip”. Rapid light curves (RLCs) were generated by exposing algal tissue to 8 incremental steps of increasing irradiance from 0–436 µM photons m−2 s−1 in 2009, 0–533 µM photons m−2 s−1 in 2010, and 0–614 µM photons m−2 s−1 in 2011, with 10 s at each light step (Saroussi & Beer, 2007). Replicate RLCs were generated in 2009 (3 RLCs per individual) and 2010 (2 RLCs per individual), and one RLC was generated for samples in 2011. Due to variation in experimental setup and PAR conditions across experiments, RLC intensities were higher than experimental PAR intensities in 2009 and 2010 and lower than experimental conditions in 2011 (Table S2). No RLCs were conducted on H. taenicola and A. amadelpha in 2012 because of time constraints. Using this approach of short illumination interval RLCs (<1 min), we were interested in relative comparisons of photophysiological performance between treatments (Enriquez & Borowitzka, 2010). Photosynthetic parameters were calculated from each RLC, and where RLCs were repeated on an individual, parameters were averaged for each individual before further statistical analyses.

Statistical analyses

To explore the effects of CO2 enrichment on growth and calcification, separate t-tests for each species compared responses between control and experimental treatments. Certain species were experimentally manipulated in multiple years; to examine overall effects on species independent of experimental year, data across years were pooled. Additional independent t-tests were run in each year for those species, because the experimental setup and sample size varied slightly from year-to-year. Prior to analysis conducted in statistical software JMP v.10, data were tested for the assumptions of normality and homogeneity of variances with the Shapiro–Wilks test and diagnostic q–q plots.

To examine photophysiological response to CO2 enrichment, the electron transport rates (ETR) from each RLC was plotted against irradiance and fit to a three parameter model (Frenette et al., 1993) to estimate the initial slope of the curve (α, µM electrons µM photons−1), the maximum relative electron transport rate (rETRMax, µM electrons m−2 s−1), and photoinhibition (β, µM electrons µM photons−1) (Platt, Gallegos & Harrison, 1980). Mean parameter estimates were averaged across samples within a treatment level for each species. In 2009 and 2010, several RLCs were generated for an individual alga; parameters were averaged within an individual before treatment effects were explored. The analyses were conducted using the software GraphPad Prism (v.6) and in all cases the model fit the data well with R2>0.90 and p < 0.001. Parameters were compared for each species between treatments using independent t-tests as described above.

Meta-analysis

Meta-analyses were used to combine data across independent experiments and to explore potential differences in functional group responses to OA. Each species was categorized as fleshy macroalgae, upright calcareous algae, or CCA. Species that became sexually reproductive during experiments (A. spicifera, A. amadelpha, C. serrulata 2011) were not included in the meta-analysis because a large portion of the algal thallus senesced, or for holocarpic species the entire thallus disintegrated, after gamete/spore release and it was not possible to differentiate between the effects of reproduction versus OA treatment on algal biomass. Species tested across multiple years were included as independent data sets, yielding 3 fleshy macroalgae, 6 upright calcareous algae, and 3 CCA representatives. A random-effects model of standardized mean differences (Cooper, Hedges & Valentine, 1994) was used to estimate within and across experiments variance components; effect size was weighted both by sample size and pooled standard deviation. A one-tailed z-test of significance (against zero) of the mean effect size of CO2 enrichment was used for algal growth and calcification responses. OA treatments were expected to enhance fleshy algal growth (H0: mean effect size ≤ 0) and decrease algal calcification (H0: mean effect size ≥ 0). There was no a priori expectation of photosynthetic responses to OA and thus a two-tailed z-test was used for the meta-analyses of photosynthetic parameters (see Supplemental Information for details).

Results

Experimental conditions

CO2 enrichment treatments effectively simulated near future seawater carbonate chemistry and OA as compared to present-day ambient air controls (Table 1). Biological activity (i.e., photosynthesis and respiration) introduced variability into carbonate chemistry conditions in both ambient and high pCO2 treatments (Table S3). Diel variability in carbonate chemistry was not characterized, however, based on previous studies photosynthesis likely caused higher pH during the day, whereas respiration reduced pH at night (Ohde & van Woesik, 1999). Discrete water samples and pH probe measurements were collected at approximately the same time of day (2000) during all experiments. The average difference (±SE) between the mean measured pHSW and the mean pHSW calculated from measurements of AT and CT was 0.1 (±0.05) (n = 32). Considering the robustness of pH probe measurements in comparison to certified Tris buffer (±0.001), the relatively small difference between measured and calculated pH, and the frequency of samples for measured pH (n = 9–17) (Table 1) versus calculated pH (n = 2–4) (Table S3), measured pHSW is the most appropriate parameter to describe differences in carbonate chemistry among experimental replicates. Most other physical conditions were consistent across years, but due to changes in experimental facilities, irradiance levels were higher and more representative of shallow reef environs in 2011 and 2012; oxygen levels were also higher in those years (Table 1).

Species–specific effects of CO2 enrichment on calcification and growth

High CO2 conditions decreased net calcification rates in 4 of the 6 calcareous species, and potentially enhanced net growth in 2 of the 5 fleshy species (Fig. 1; Table 2). CO2 enrichment significantly decreased calcification in the red calcareous macroalga D. marginata (by 98%), and the two CCA Lithophyllum sp. (by 185%) and T. prototypum (by 190%) relative to controls (Table 2). The response of the green calcareous algae in the genus Halimeda was species–specific: the effect of CO2 enrichment on net calcification rates was negative for H. opuntia (when repeated experiments were pooled) but negligible for H. taenicola (Table 2). CO2 enrichment significantly increased growth in the fleshy red macroalga H. pannosa (by 93%) relative to controls (Table 2). The fleshy brown macroalga D. bartayresiana showed slight but non-significant increases in growth in high CO2 likely due to small sample size and lack of power (β = 0.46; Table 2).

Table 2 Results of pooled growth and photosynthetic parameters in response to treatment conditions.

The results of independent t-tests to analyze the effect of CO2 enrichment on response variables for each species. Responses of species used in multiple experiments (different years) were pooled and averaged across years by treatment to calculate an overall mean for each species. CO2 treatment was treated as a fixed, independent factor. Degrees of freedom (df) are the same for all photosynthetic parameters. Each experimental replicate (n) consisted of one aquarium containing one algal individual. Statistically significant differences (p < 0.05) are emphasized in bold.

	Growth	rETRMax	α	β	
Species	df	t	p	df	t	p	t	p	t	p	
Fleshy macroalgae	
A. spicifera	10	1.15	0.275	9	2.55	0.031	0.341	0.741	0.088	0.932	
A. amadelpha	18	3.12	0.006								
C. serrulata	22	0.066	0.948	19	0.282	0.781	0.350	0.730	0.356	0.726	
D. bartayresiana	8	2.13	0.066	8	1.55	0.159	0.274	0.791	1.14	0.292	
H. pannosa	5	4.90	0.004	5	0.186	0.556	0.602	0.60	0.624	0.560	
Upright calcareous algae	
D. marginata	8	3.83	0.005	8	0.440	0.83	0.092	0.929	0.823	0.434	
G. rugosa	10	1.63	0.134	10	4.10	0.002	1.71	0.118	0.760	0.465	
H. opuntia	16	2.59	0.020	16	0.046	0.964	1.43	0.171	0.223	0.827	
H. taenicola	38	0.21	0.832	18	0.193	0.849	2.23	0.039	1.62	0.123	
Crustose coralline algae	
Lithophyllum sp.	16	5.28	<0.0001	16	0.582	0.569	0.280	0.783	1.0	0.332	
L. prototypum	6	2.79	0.032	6	0.357	0.733	0.404	0.700			

In addition to across species variability in the growth response, there was intra-specific variation to CO2 enrichment across different years of experiments (Fig. 2). The trends and absolute magnitude in growth responses remained the same for 2 of the 4 species tested over multiple years. Irrespective of year, the calcareous green alga H. opuntia calcified significantly less (by 14.55 mg g−1 d−1 in 2009 and 12.97 mg g−1 d−1 in 2011) under high CO2 conditions (Table 3), although the relative response varied by year. H. opuntia calcified 200% less at high CO2 than ambient conditions and even experienced net dissolution in 2009, but only calcified 50% less in 2011 and experienced net growth, despite the same high CO2 conditions (Table S4). Lithophyllum sp. showed a consistent response to CO2 treatment in direction and absolute and relative magnitude across years. Lithophyllum sp. calcified 185% less at high CO2 in both 2009 and 2011 (Table S4). H. taenicola calcified 89% less at high CO2 relative to controls in 2009, but there was no significant difference in calcification during the 2010 and 2012 experiments (Table 3). C. serrulata grew significantly more at high CO2 in the 2010 experiment, however in 2011 C. serrulata grew less in the CO2 enrichment treatment than in ambient conditions (Table 3).

Figure 2 Species–specific growth response to treatment conditions.

The mean (±SE) change in either fleshy or calcareous biomass following exposure to either present-day ambient air controls (open circles) or predicted OA treatments (closed circles) for species tested in multiple experiments. The dashed line is positioned at zero to indicate relative growth or loss of tissue for (A) Halimeda opuntia, (B) Halimeda taenicola, (C) Lithophyllum sp., and (D) Caulerpa serrulata. Fleshy macroalgae are shown in brown, upright calcareous algae in green, and CCA in red. * Indicates a significant difference between treatments as determined by independent t-tests (results reported in Table 3).

Several fleshy macroalgal species became reproductive in CO2 treatments over the course of our study, as evidenced by the presence of fertile tissue which eventually released gametes or spores leaving behind only a small portion of the vegetative thallus. All samples of A. spicifera and A. amadelpha released spores or gametes, respectively, upon exposure to treatment conditions. In 2011, C. serrulata also reproduced, causing tissue loss in both ambient and CO2 treatments; 40% of Caulerpa individuals in the ambient treatment reproduced, whereas 100% of Caulerpa samples in the CO2 enrichment treatments reproduced.

Species–specific effects of CO2 enrichment on photophysiology

Exposure to CO2 treatments had no detectable effect on relative photophysiology of the 9 species tested, with a few exceptions (Fig. 3). CO2 enrichment significantly increased the maximum photosynthetic capacity (rETRMax) in the calcareous red alga G. rugosa (Fig. 3A, Table 2) relative to the control. In the fleshy red alga A. spicifera, rETRMax was significantly lower following exposure to high CO2, however, these individuals had reproduced during the experiment and the remaining vegetative tissue following gamete release was not representative of healthy algal tissue. In the calcareous green alga H. taenicola, the initial slope of the RLC (α) was significantly depressed after CO2 enrichment (Fig. 3B, Table 2). There was no evidence of photoinhibition (β) in any of the species tested (Fig. 3C, Table 2).

Figure 3 Photosynthetic response of fleshy and calcareous algae to treatment conditions.

The mean (±SE) photosynthetic parameters from RLCs following exposure to either present-day ambient air controls (open circles) or predicted OA treatments (closed circles). Species tested in multiple experiments were pooled across years. RLCs were measured fluorometrically with a pulse amplitude modulated fluorometer (PAM), and fit to the model of Platt, Gallegos & Harrison (1980). From the model we derived (A) maximum photosynthetic performance (rETRMax), (B) photosynthetic efficiency (α), and (C) photoinhibition (β). Parameters from replicate RLCs were averaged for each individual. Fleshy macroalgae are shown in brown, upright calcareous algae in green, and CCA in red. * Indicates a significant difference between treatments as determined by independent t-tests (results reported in Table 2).

Table 3 Results of growth/calcification by species and year.

The mean growth and calcification rates of species tested in multiple experiments were examined using independent t-tests for each species by year; CO2 treatment was treated as a fixed, independent factor. Each experimental replicate (n) consisted of one aquarium containing one algal individual. Statistically significant differences (p < 0.05) are emphasized in bold.

		Growth	
Species	Year	df	t	p	
Fleshy macroalgae	
C. serrulata	2010	10	4.28	0.002	
	2011	8	1.75	0.119	
Upright calcareous algae	
H. opuntia	2009	6	7.32	0.0003	
	2011	8	3.62	0.007	
H. taenicola	2009	6	5.93	0.001	
	2010	10	0.224	0.827	
	2012	18	0.612	0.548	
Crustose coralline algae	
Lithophyllum sp.	2009	6	4.10	0.006	
	2011	8	3.43	0.009	

Meta-analysis of experiments across years

Experimental effects were combined across species to assess the consistency of physiological responses to CO2 enrichment within different algal functional groups using meta-analyses. The mean effect size for calcification and growth was significantly greater than zero for fleshy species, but significantly less than zero for both groups (upright and encrusting) of calcareous species (Table 4; Fig. 4A). There was no overall effect of CO2 enrichment on photophysiology (rETRMax, α, β) relative to the control for algal functional groups (Fig. 4, Table 4). The variation between experiments was never significantly different from 0 (Q ≤ 2.04, p > 0.05 for each functional group and response variable; Table 4), indicating that the inconsistencies in PAR did not influence the overall response of fleshy versus calcareous algae to OA. Due to the significant effect of CO2 enrichment on growth and calcification rates across experiments, and the lack of significant variation in the strength of this response, we pooled species across years to show overall trends in treatment responses (Fig. 1).

Figure 4 Functional group responses to OA.

Mean (± 95% CI) effect sizes were calculated to explore the cumulative effects of OA on algae categorized into functional groups (fleshy macroalgae, upright calcareous algae, and crustose coralline algae (CCA). Species that reproduced during experiments were not included in this analysis. The dashed line is positioned at zero to indicate a relative increase or decrease following exposure to OA conditions for (A) change in weight, (B) maximum photosynthetic capacity (rETRMax), (C) photosynthetic efficiency (α), and (D) photoinhibition (β). Fleshy macroalgae are shown in brown circles, upright calcareous algae in green, and CCA in red. * Indicates an effect size different than zero as determined by meta-analysis (results reported in Table 4).

Table 4 Meta-analysis results.

Heterogeneity (QT) in overall analyses and results from a random effects model of standardized mean differences for response variables pooled by functional group: fleshy macroalgae, upright calcareous algae, or crustose coralline algae (CCA). Statistically significant values (p < 0.05) are emphasized in bold. rETRMax, maximum relative electron transport rate (µM photon m−2 s−1); α, photosynthetic efficiency or initial slope of the rapid light curve (µM electrons µM photons−1); β, photoinhibition (µM electrons µM photons−1).

Response	df	QT	p	k	Mean effect size	Z	p	
Fleshy macroalgae	
Growth	19	0.07	>0.05	3	16.1 ± 12.5	2.11	0.017	
rETRMax	12	0.18	>0.05	3	0.454 ± 6.19	0.144	0.886	
α	12	0.07	>0.05	3	−0.005 ± 0.13	0.073	0.471	
β	12	0.16	>0.05	3	−0.0004 ± 0.0009	0.632	0.2248	
Upright calcareous algae	
Growth	4	2.04	>0.05	7	−10.8 ± 4.7	3.80	0.0001	
rETRMax	4	1.62	>0.05	6	0.031 ± 3.56	0.017	0.987	
A	4	0.72	>0.05	6	−0.020 ± 0.06	0.722	0.470	
B	4	0.60	>0.05	6	0.001 ± 0.01	0.756	0.4333	
Crustose coralline algae	
Growth	6	0.08	>0.05	3	−0.405 ± 0.35	1.90	0.029	
rETRMax	6	0	>0.05	3	0.693 ± 27.4	0.339	0.735	
α	6	0.05	>0.05	3	−0.002 ± 0.09	0.053	0.941	
β	6	0	>0.05	3	−0.001 ± 0.004	0.169	0.2637	

Discussion

This series of experimental manipulations indicate that tropical algae respond differently to CO2 enrichment depending on species and whether or not they are calcified. When combining data from multiple experiments, calcareous algae experienced a reduction in biomineralization while fleshy algae became more productive. The magnitude of algal growth and calcification responses to OA conditions varied by species, and occasionally, within a species over multiple experiments. In contrast, there was no effect of CO2 enrichment on algal photophysiology relative to controls as measured by short illumination RLCs. Furthermore, exposure to OA conditions initiated sexual reproduction in 3 out of 5 species of fleshy macroalgae tested. These results support the hypothesis that OA has differential effects on the growth of fleshy macroalgae and the calcification of calcareous algae.

Biomineralization by seaweeds substantially contributes to carbonate production on tropical reefs and these results suggest that OA may decrease reef formation and cementation services provided by these often over-looked ecosystem engineers. In these experiments, OA decreased calcification of calcareous green algae (H. opuntia and H. taenicola) and caused net dissolution of calcareous red macrophytes and CCA (D. marginata, G. rugosa, Lithophyllum sp., and L. prototypum). Many other studies have reported decreased calcification as a consequence of simulated OA for tropical (Table 5) and temperate calcareous algae even in milder acidification scenarios than used in our study (see Koch et al., 2013 for review). However, much of the previous work exploring OA effects on calcareous algae across ecosystems has focused on the crustose coralline algae (family Corallinaceae) and this study is among the first to expand to different taxonomic entities such as the lightly calcified red algae D. marginata and G. rugosa (Table 5).

Table 5 OA effects on tropical benthic macroalgae.

A summary of findings to date from experiments exploring OA effects on growth, calcification, and photosynthesis in tropical benthic macroalgae. Only business-as-usual OA experiments (800–1200 µatm) are included. +, positive effect; −, negative effect; 0, no effect.

Species	Growth/
Calcification	Photosynthesis	Reproduction	Reference	
Fleshy macroalgae	
Acanthophora spicifera	0	−	+	This study	
Avrainvillea amadelpha	−		+	This study	
Caulerpa serrulata 2010	+	0		This study	
Caulerpa serrulata 2011	0	0	+	This study	
Dictyota bartayresiana	+	0		This study	
Hypnea pannosa	+	0		This study	
Lobophora papenfussii	−			(Diaz-Pulido et al., 2011)	
Upright calcareous algae	
Galaxaura rugosa	0	+		This study	
Dichotomaria marginata	−	0		This study	
Halimeda opuntia	−	0		This study	
Halimeda taenicola	0	0		This study	
Halimeda cylindracea	−	−		(Sinutok et al., 2011; Sinutok et al., 2012)	
Halimeda macroloba	−	−		(Sinutok et al., 2011; Sinutok et al., 2012)	
Halimeda incrassata	+			(Ries, Cohen & McCorkle, 2009)	
Crustose coralline algae	
Lithophyllum prototypum	−	0		This study	
Lithophyllum sp.	−	0		This study	
Hydrolithon sp.	−	+		(Semesi, Kangwe & Bjork, 2009)	
Porolithon onkodes	−	−		(Anthony et al., 2008; Diaz-Pulido et al., 2012; Johnson & Carpenter, 2012; Comeau, Carpenter & Edmunds, 2012; Comeau et al., 2013)	
Neogoniolithon sp.	+			(Ries, Cohen & McCorkle, 2009)	
Mixed CCA	−			(Jokiel et al., 2008; Kuffner et al., 2008)	

The results of this study indicate that calcareous algae calcified less after two weeks of exposure to CO2 enrichment than ambient controls, but the response varied by functional group. CCA, which deposit the more soluble high Mg-calcite (12–18% MgCO3; Milliman, Gastner & Muller, 1971), experienced net dissolution in the OA treatments, where ΩMg-calcite was ≤1 (using the solubility constant estimated by Lueker, Dickson & Keeling, 2000), despite assuming our samples deposited the conservative lower range of 8% Mg mole fraction. Intracellular dolomite (CaMg[CO3]2), a stable form of carbonate, can be the source of Mg in other species of CCA and actually reduces net thallus dissolution at higher skeletal mole fractions (Nash et al., 2012). The exact mineral composition of the carbonate in our CCA species is unknown, but was not robust to our treatment conditions. The calcareous upright algae all deposit aragonite and calcified less under OA, but only experienced net dissolution in one instance. Differences in the magnitude of effects between calcareous species may be influenced by species–specific mechanisms of calcification (Price et al., 2011; Comeau, Carpenter & Edmunds, 2012; Koch et al., 2013), mineralogy of CaCO3 deposited (Ries, Cohen & McCorkle, 2009), and potential compensatory or antagonistic effects of high CO2 on photosynthesis (Table 5). Differences in within-species susceptibility to OA demonstrate the complexity of how ocean acidification may influence biological and chemical interactions in tropical marine primary producers. Within-species responses across years of experiments may have been driven by changes in dissolution versus calcification or by net growth rate, and the relative contribution of dissolution versus calcification in influencing net effects of OA on organisms should be a focus in future studies.

Understanding the effects of OA on algal physiology is difficult because photosynthesis and calcification are inextricably coupled. In the process of fixing carbon, algal photosynthesis alters the intracellular environment in favor of CaCO3 precipitation (Borowitzka & Larkum, 1976). In the external environment, photosynthesis also has the potential to alter carbonate chemistry and to create conditions more favorable for calcification (Gattuso, Pichon & Frankignoulle, 1995; Anthony, Kleypas & Gattuso, 2011; Smith et al., 2013). Fleshy macroalgae that are currently carbon limited are hypothesized to be affected positively by increasing CO2 concentrations (Gao et al., 1991), which is demonstrated here, but these effects are species and condition specific. Previous studies have documented both positive and negative effects of CO2 enrichment on growth in fleshy macroalgae (Table 5). Enhanced algal growth also has been documented in situ in ecosystems near underwater volcanic vents where conditions of low pH and high CO2 facilitate communities dominated by fleshy organisms (Hall-Spencer et al., 2008; Fabricius et al., 2011). It has been hypothesized that higher concentrations of dissolved CO2 would enhance fleshy macroalgal growth by stimulating photosynthesis. However, despite the fact that fleshy algae grew more with high CO2 there was not a concurrent response in photosynthetic parameters measured from chlorophyll fluorescence. While the fluorescence technique is used widely to monitor algal photophysiology, it can be highly variable (Edwards & Kim, 2010) and provides only an instantaneous snapshot of photophysiological function. Short illumination RLCs (<1 min) are not comparable to estimates obtained using oxygen evolution from photosynthesis-irradiance curves because there is not sufficient time with RLCs for organisms to reach steady-state flow of electrons (Enriquez & Borowitzka, 2010). Thus, it may not be the most suitable technique to assess the cumulative effects of CO2 enrichment on algal photophysiology and more direct measures of photosynthesis are preferred.

Predicting the response of primary producers to high CO2 is complex and may depend on resource acquisition strategies that are species–specific and potentially plastic over time. The primary substrate for the photosynthetic enzyme Rubisco in all marine algae is dissolved CO2. Seaweeds must compensate for the slow rates of CO2 diffusion through seawater, as opposed to air, as well as the higher concentration of HCO3− compared to CO2. Some primary producers have developed carbon concentrating mechanisms (CCM) that increase the concentration of CO2 in the proximity of Rubisco (Raven, 1970). Thus, the presence or absence of CCMs may influence species–specific responses to CO2 enrichment (Hurd et al., 2009; Koch et al., 2013), and changes to CCM activity levels may explain the mixed responses of photosynthesis in the literature, as well as the growth results documented here. One possible mechanism that may have facilitated increased algal productivity under high CO2 in the present study, without concurrent increases in rETRMax or α, may have been an increase in algal energy reserves through down regulation of energetically costly CCMs, noted in another tropical green macroalga (Liu, Xu & Gao, 2012) and phytoplankton (Eberlein, Van de Waal & Rost, in press). An additional alternative hypothesis is that nitrate reductase activity, an enzyme that reduces nitrate to nitrite, can be stimulated by CO2 (Hofmann, Straub & Bischof, 2013), potentially releasing seaweed from nitrogen resource limitation in oligotrophic coral reef ecosystems. Furthermore, photophysiology should be assessed using more direct techniques in addition to RLCs such as measuring oxygen evolution rates, in order to accurately quantify photosynthetic rates. Predicting changes in enzymatic activity is critical to understanding mechanisms behind species–specific responses to OA, yet basic physiological descriptions are lacking for the majority of tropical algae, including the species used in the present study.

This and other studies have documented high variability among species in response to OA. However, there also was within species variability across years, suggesting that species–specific responses to OA may be context dependent. For example, due to logistical constraints experiments conducted in 2009 and 2010 had substantially lower daily mean irradiances than in 2011 and 2012 (ESM Table 2). Although mixing rates were consistent from year to year, flow rates in experimental aquariums were relatively low. Thus, care should be taken when extrapolating these biological responses to OA under higher water flow regimes. Few studies have experimentally tested the effects of both water flow and OA on coral reef algae, although flow rate has been shown to be an important factor influencing pH gradients within the diffusive boundary layer (DBL) (Hurd et al., 2011) and the response of some reef calcifiers to high CO2 (Anthony et al., 2013). Increasing DBL thickness, with decreasing water flow, may buffer organisms against changes in the carbonate chemistry of bulk seawater by providing a metabolically mediated microenvironment of higher pH within the DBL. Furthermore, the biological variability in carbonate conditions introduced by algal photosynthesis and respiration in the contained, aerated volume of water likely created a diel cycle in pH that may have approximated carbonate chemistry variability on a shallow reef flat (Hofmann et al., 2011). Variability in pH conditions has been shown to influence growth rates of coralline algae (Johnson, Moriarty & Carpenter, 2014), therefore care should be taken when extrapolating the results from the present study to other systems. Diel cycles in carbonate chemistry were not characterized in this experiment, and have been infrequently included in descriptions of experimental conditions in many OA studies. However, the variability in all experimental conditions across this suite of experiments is far less than that of experiments combined in several recent meta-analyses (Hendriks, Duarte & Alvarez, 2010; Kroeker et al., 2010; Kroeker et al., 2013)). With the meta-analysis approach used here to explore effects of OA across experiments, we accounted for the variability within and across experiments and found that OA treatment was a significant driver of enhanced growth in fleshy macroalgae, and loss of calcified biomass in upright calcareous algae and CCA.

Some within-species variability in response to OA treatment was due to the induction of sexual reproduction following exposure to treatment conditions. Higher irradiance levels can modulate the negative effects of high CO2 on algal responses to OA (Sarker et al., 2013; Yildiz et al., 2013), including potentially triggering reproduction and may explain the inconsistent results from year to year, specifically for C. serrulata. In 2011 the decrease in C. serrulata growth likely was a result of the loss of algal tissue in individuals that became sexually reproductive upon exposure to high CO2 and relatively higher irradiance. Similar reproductive responses to treatment conditions were also noted for related A. amadelpha and for a red macroalga, suggesting that there may be an interactive effect between irradiance levels and CO2 concentrations. Sexual reproduction in these taxa has been observed in Hawaii and the Caribbean during the spring (Clifton & Clifton, 1999; Smith, Hunter & Smith, 2002). In all of these species, a large portion of the algal thallus senesced after sexual reproduction, and for the green algae the reproduction was clearly visible due to the loss of pigmentation following release of heavily pigmented gametes or spores (Clifton & Clifton, 1999). The typical progression of sexual reproduction in Bryopsidales ranges from 1–2 days (Clifton & Clifton, 1999), and the specimens did not show signs of reproduction prior to the experiment. Furthermore, gametogenesis in C. serrulata has been shown to be induced either as a coping mechanism (Williamson, 2010) or to maximize favorable conditions (Brawley & Johnson, 1992). Experiments were conducted outside the potential seasonal reproductive cycle of tropical algae, and there was no evidence of sexual reproduction before experiments, thus it is likely that sexual reproduction was induced by experimental conditions. An alternative explanation is that the reproductive response may have been an artifact of experimental manipulation and stress associated with rapid exposure to high pCO2. The rate of exposure to high pCO2 has been shown to be an important determinant in the response of coralline algae to CO2 enrichment (Kamenos et al., 2013). Future work should explore the effects of both rate and magnitude of CO2 enrichment on reproduction in fleshy macroalgae.

OA poses an ever-increasing global threat (Kleypas et al., 1999; Hoegh-Guldberg et al., 2007) to the ecological balance and stability of tropical reef systems via disparate effects on calcareous versus fleshy taxa (Hall-Spencer et al., 2008; Fabricius et al., 2011; Porzio, Buia & Hall-Spencer, 2011). It is difficult to predict the specific responses of macroalgal taxa to CO2 enrichment; however, the patterns of response presented here suggest that growth of fleshy macroalgae on coral reefs may be stimulated by OA, while calcareous species may be depressed. Given that numerous other human impacts (overfishing, pollution, warming) negatively affect corals and other calcifying reef builders while enhancing the abundance of fleshy algae, our results suggest that OA may potentially exacerbate community shifts towards fleshy macroalgal dominated states. However, little is known about how reef species or communities will respond to the interactive effects of multiple stressors including OA. Given the importance of coral reefs for supporting biodiversity (Knowlton, 2001), as well as human populations and economies of coastal nations (Moberg & Folke, 1999), it is imperative that we understand the scope of species responses to impending rapid climate change.

Supplemental information

Table S1 Taxonomy and functional groupings of specimens

The taxonomy and functional groupings of the algae used in CO2 experiments from 2009–2012. Sample size (n) is the number of replicates within a treatment per experiment, where each aquarium contained one alga.

Click here for additional data file.

Table S2 Experimental design of CO2 enrichment experiments

The design of CO2 enrichment experiments conducted on Palmyra Atoll from 2009–2012. In each experiment, handheld meters were used to monitor conditions, water baths to regulate temperature, and shading to reduce photo-inhibition, but the instruments and approaches varied by experiment. PAR: photosynthetically active radiation (µM photons m−2 s−1).

Click here for additional data file.

Table S3 Carbonate chemistry parameters for experiments conducted on Palmyra Atoll from 2009–2012

The mean (±SE) carbonate system parameters of CO2 enrichment experiments conducted on Palmyra Atoll from 2009–2012. Water samples were collected at the same time of day (in duplicate after 2009) from a subset of blank control (no biological material), ambient air, or high pCO2 aquariums every 2–3 days; all analyses were conducted at 20 °C. Means were calculated by averaging control and treatment water samples over the duration of the experiment (only control samples were collected in 2009). AT, total alkalinity, CT, total inorganic carbon, pHSW, pH seawater scale, pCO2, partial pressure of CO2, ΩAr, saturation state of aragonite, ΩCa, saturation state of calcite.

Click here for additional data file.

Table S4 Growth rate of fleshy and calcareous algae

The mean (±SE) growth rate for each species across experiments standardized to the initial weight and the duration of the experiment. Sample size (n) is the number of replicates within a treatment per experiment.

Click here for additional data file.

Table S5 Photosynthetic parameters in response to treatment conditions

The mean (±SE) photosynthetic parameters across species in response to CO2 enrichment as estimated from rapid light curves with the equations of Platt, Gallegos & Harrison (1980). Sample size (n) is the number of replicate samples within a treatment, and repeats is the number of RLCs run for each sample. rETRMax, maximum relative electron transport rate (µM photon m−2 sec−1); α, photosynthetic efficiency or initial slope of the rapid light curve (µM electrons µM photons−1); β, photoinhibition µM electrons µM photons−1).

Click here for additional data file.

Supplemental Information 1 Details of meta-analysis methodology

Details of the statistical analysis involved with the meta-analysis.

Click here for additional data file.

We thank the US Fish and Wildlife Service (USFWS) and The Nature Conservancy for granting access to Palmyra Atoll and providing logistical support. The A Dickson lab provided chemical analyses and A Meyer, K Furby, J Harris, S Hamilton, S Sandin, J Tootell, and B Zgliczynski provided logistical field support. Scripps Institution of Oceanography (SIO) is a member of the Palmyra Atoll Research Consortium (PARC); this is contribution number PARC-0107.

Additional Information and Declarations

Competing Interests

Author Contributions

Field Study Permissions

The authors declare there are no competing interests.

Maggie Dorothy Johnson and Nichole N. Price conceived and designed the experiments, performed the experiments, analyzed the data, wrote the paper, prepared figures and/or tables, reviewed drafts of the paper.

Jennifer E. Smith conceived and designed the experiments, performed the experiments, contributed reagents/materials/analysis tools, wrote the paper, reviewed drafts of the paper.

The following information was supplied relating to field study approvals (i.e., approving body and any reference numbers):

This research was conducted under the USFWS special use permits 12533-09020, 12533-12011, and 12533-12012.

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
