# Peer review of "Contrasting effects of ocean acidification on tropical fleshy and calcareous algae"

_PeerJ, doi:10.7717/peerj.411_

## Round 0.1 · original submission · Minor Revisions

This is a nice piece of work on a group that typically receives less attention than the scleractinian corals, and I enjoyed reading it. I agree with the evaluations of the referees that this work is of interest and value to the community, and that the manuscript is already solid. I also appreciate the suggestions of the referees that there are a couple issues which would improve the manuscript. The primary issue that I share with both referees is regarding clarity of the experimental design and analyses which must be corrected before the manuscript is published. I believe that the authors can resolve these issues relatively easily, and the referees provide good feedback on where revisions to the text would be particularly helpful. I look forward to seeing the revised manuscript.

Reviewer 1 ·

Basic reporting

To clarify the details of the experimental comparisons it would be helpful to provide species acronyms in Table 1, so it is readily apparent from the main body of the paper which taxa are linked to which experiments.

Experimental design

The authors state “Shorter experiments (2 weeks) are merited because Palmyra provides a unique setting for global change experiments due to the absence of potentially confounding local anthropogenic impacts.” The time frame of the experiment is not necessarily merited by the fact that Palmyra is a unique setting. It could equally well be that organisms in a heavily impacted area would respond more strongly to OA assaults due to multiple stressors and would be well matched to short incubations.

Was there any flow in the glass jars? If not, it is necessary to discuss the implications for the response variables in a low to no flow condition.

Line 210: As light and CO2 level varied greatly between experiments, the justification for pooling across years is not clear.

The authors state that light data was recorded with Hobo loggers, which only measure light in lux or lumens/ft2. Light is reported however in µmol photons m-2 sec-1.

No light values are reported for the intensities used in the RLCs. What intensities were used and how did this compare to the experiment and the field light levels? Is there any correlation between light levels and photophysiological performance that may explain difference among taxa responses?

How rapid were the light curves? If used on automated settings, it is often not enough time for the organisms to reach the steady-state or steady flow of electrons between PSII and PSI, and as such do not represent photosynthetic rates or photoacclimation. The RLC data may still be useful in the comparative framework, but care should be given to the interpretation. see Borowitzka and Enriquez (2010) The Use of the Fluorescence Signal in Studies of Seagrasses and Macroalgae. This may influence your discussion on lines 357-361.

Validity of the findings

While the details of the experimental design need further clarification, it appears the authors’ conclusions are supported by the results and are based on appropriate, although not cutting edge, experiments. With attention to the experimental design concerns listed above, this manuscript will have suitable support to assess the validity of the conclusions.

Additional comments

Johnson and coauthors present a large experimental effort to address the timely hypotheses that there will be differential responses between algal taxa under future OA conditions; specifically that OA reduces net calcification of calcareous algae and stimulates photosynthesis and growth in fleshy algae. They conducted several OA experiments between 2009 and 2012 on multiple algal taxa across fleshy, upright calcified and crustose coralline functional groups. While the experiments are varied and the experimental detail still needs a bit of clarification, the meta-analytical approach is a robust way to test for responses across differing experiments giving weight to the authors’ conclusions.

Reviewer 2 ·

Basic reporting

Overall the manuscript is well-written and appropriately structured. I've included several specific suggestions as comments in the PDF. Several of these include foods for thought or alternative interpretations of the data which are worth considering, and possibly including in a modified manuscript.

Experimental design

For the most part the design and methods appear appropriate, though there are a few spots where the methods and chemistry data in particular need to be clarified in order to fully evaluate the adequacy of the data. These are discussed as detailed comments in the PDF.

Validity of the findings

I'd like to see more information and greater clarity on the chemistry information. However, assuming that these concerns can be adequately addressed, the findings and conclusions follow logically. An appropriate repository for the data would be something like the website http://pangaea.de.

Additional comments

Overall this seems like a nice paper to me and potentially an important contribution to the field. My major criticism is that I feel the chemistry information provided is not explicit or detailed enough to be able to rigorously evaluate the data. I like the replication across years and the meta-analysis approach. The data pretty clearly show that there is some real variation for the same species of algae exposed to OA for reasons that are not really understood. The same is true for most other organisms when we look at OA effects. Rather than shying away from this variation, I think it is important for all of us to focus on it (and the authors point it out) because that is probably where the action is going to lie in terms of figuring the major issues regarding OA effects on marine organisms. Why does organism X react strongly to OA when we test it once, and react only weakly or not at all the next time? We have ideas on what is driving some of this variation for some organisms, but it looks like the effects of OA really can vary a lot for the same organisms, even if there are some general trends, as suggested by the meta-analysis here. If the experimental conditions (= chemistry) can be detailed more clearly, and perhaps after considering some of the pieces of food for thought I've suggested in the PDF, I see the manuscript as highly publishable.

Annotated reviews are not available for download in order to protect the identity of reviewers who chose to remain anonymous.

---

## Round 0.2 · accepted · Accept

I have now read through your manuscript and your responses to the reviewers, which I feel is well considered. I appreciate your incorporation of the feedback to the manuscript and am happy to accept your revised manuscript.